# Comparison of Direct Oral Anticoagulant Use for the Treatment of Non-Valvular Atrial Fibrillation in Pivotal Clinical Trials vs. the Real-World Setting: A Population-Based Study from Southern Italy

**DOI:** 10.3390/ph14040290

**Published:** 2021-03-25

**Authors:** Ylenia Ingrasciotta, Andrea Fontana, Anna Mancuso, Valentina Ientile, Janet Sultana, Ilaria Uomo, Maurizio Pastorello, Paolo Calabrò, Giuseppe Andò, Gianluca Trifirò

**Affiliations:** 1Department of Biomedical and Dental Sciences and Morpho-Functional Imaging, University of Messina, 98158 Messina, Italy; anna.mancuso.2101@gmail.com (A.M.); vientile@unime.it (V.I.); jaysultana@unime.it (J.S.); 2Unit of Biostatistics, Fondazione IRCCS Casa Sollievo della Sofferenza, San Giovanni Rotondo, 71013 Foggia, Italy; a.fontana@operapadrepio.it; 3Department of Pharmacy, Palermo Local Health Unit, 90100 Palermo, Italy; ilariauomo@gmail.com (I.U.); dipfarmaceuticoasppa@gmail.com (M.P.); 4Division of Cardiology, A.O.R.N. “Sant’Anna e San Sebastiano”, 81100 Caserta, Italy; paolo.calabro@unicampania.it; 5Department of Translational Medical Sciences, University of Campania “Luigi Vanvitelli”, 81100 Naples, Italy; 6Department of Clinical and Experimental Medicine, Section of Cardiology, University of Messina, 98158 Messina, Italy; giuseppe.ando@unime.it; 7Department of Diagnostic Public Health, University of Verona, 37129 Verona, Italy; gianluca.trifiro@univr.it

**Keywords:** direct oral anticoagulants, nonvalvular atrial fibrillation, external validity, real world

## Abstract

Patients enrolled into pivotal randomized controlled trials (RCTs) may differ substantially from those treated in a real-world (RW) setting, which may result in a different benefit–risk profile. The aim of the study was to assess the external validity of pivotal RCT findings concerning direct oral anticoagulants (DOACs) for the treatment of nonvalvular atrial fibrillation (NVAF) by comparing patients recruited in RCTs to those treated with DOACs registered in a southern Italian local health unit (LHU) in the years 2013–2017. The Palermo LHU claims database was used to describe the baseline characteristics of incident DOAC users (washout > 1 year) with NVAF compared with those of enrolled patients in DOAC pivotal RCTs. In the RW, DOAC treatment discontinuation was calculated during the follow-up and compared with DOAC treatment discontinuation of enrolled patients in DOAC pivotal RCTs. Rates of effectiveness and safety outcomes during the follow-up were calculated in an unmatched and in a simulated RCT population, by matching individual incidental RW and RCT DOAC users (excluding edoxaban users) on age, sex, and CHADS_2_ score. Overall, 42,336 and 7092 incident DOAC users with NVAF were identified from pivotal RCTs and from the RW setting, respectively. In RCTs, DOAC use was more common among males (62.6%) compared with an almost equal sex distribution in the RW. RCT patients were younger (mean age ± standard deviation: 70.7 ± 9.2 years) than RW patients (76.0 ± 8.6 years). Compared with RCTs, a higher proportion of RW dabigatran users (30.4% vs. 19.6%) and a lower proportion of RW apixaban (15.9% vs. 25.3%) and rivaroxaban (20.4% vs. 23.7%) users discontinued the treatment during the follow-up (*p*-value < 0.001). The rate of ischemic stroke was lower in RW high-dose dabigatran users (unmatched/-matched population: 0.40–0.11% per year) than in the Randomized Evaluation of Long-Term Anticoagulation Therapy (RE-LY) population (0.93% per year). Major bleeding rates were lower in RW users than in RCT users. In conclusion, except for dabigatran, a lower proportion of DOAC discontinuers was observed in the real-world than in pivotal RCT settings. This study provides reassurance to practicing physicians that DOAC use appears to be effective in stroke prevention and is likely safer in RW patients than in RCT enrolled patients. These results may be related to a lower burden of comorbidities despite more advanced age in the RW population compared to the pivotal RCT population.

## 1. Introduction

Atrial fibrillation (AF) is the most common sustained arrhythmia, predisposing patients to a higher risk of thromboembolic events and mortality [1]. In Italy, in the last decade, the prevalence of AF in the general adult population was 1.9%, increasing with age (0.1–0.2% in persons aged < 49 years versus 10–17% in persons aged ≥ 80 years old) [2].

For several years, vitamin K antagonists (VKAs) such as warfarin were the only drugs approved for stroke prevention in AF patients. The Italian Drug Agency (AIFA) approved the first direct oral anticoagulants (DOACs), dabigatran, rivaroxaban, and apixaban, for the prevention of stroke in adult patients with nonvalvular atrial fibrillation (NVAF) with one or more risk factors (prior stroke/transient ischemic attack (TIA), age ≥ 75 years, heart failure, diabetes or hypertension) in 2013 [3,4,5]. Edoxaban was approved in Italy in 2016 for the same indication [6].

In comparison to VKAs, DOACs offer several advantages, such as rapid onset and end of pharmacological action and fewer drug–drug and drug–food interactions. In addition, DOACs do not require periodic international normalized ratio (INR) monitoring and dose adjustments, thereby simplifying treatment management. More importantly, as demonstrated by pivotal trials [7,8,9], rivaroxaban, apixaban, and high-dose dabigatran were superior to warfarin in terms of efficacy, while high- and low-dose edoxaban [10] and low-dose dabigatran were found to be noninferior to warfarin. On the other hand, a higher risk of gastrointestinal bleeding was observed for all DOACs, except for apixaban, as compared to warfarin.

Although randomized controlled trials (RCTs) are considered the “gold standard” for generating evidence, especially on drug efficacy, these studies generally suffer from limited external validity [11] because patients at high risk of adverse effects, those with multiple comorbidities, or frail elderly persons are often excluded, suggesting the need for the reevaluation of the benefit–risk profile of the drugs in real-world settings. Concerning the DOACs’ pivotal RCTs specifically, these had a duration ranging from 13 to 54 months. However, it is known that DOACs are used for much longer periods in clinical practice. Moreover, pivotal RCTs applied strict inclusion criteria (e.g., age, evidence of NVAF from electrocardiogram, and at least one risk factor for bleeding) and exclusion criteria (e.g., pregnancy, conditions associated with an increased risk of bleeding, creatinine clearance of less than 30 mL/min) for patient enrolment [7,8,9,10]. This manner of selecting RCT patients potentially excludes patients who may be eligible for DOAC therapy in a real-world (RW) setting [12]. Therefore, it is important to explore if NVAF patients included in pivotal trials of DOACs are comparable to those who are treated in a RW setting and if the findings from RCTs can be ultimately generalized to RW NVAF patients.

Generally, RW data strengthen guideline recommendations and demonstrate the benefits of DOACs for stroke prevention in NVAF, with important advantages over VKA in terms of mortality and major bleeding [13]. RW data, from a clinical standpoint, may also support specific DOAC selection according to patients’ characteristics, as RCTs directly comparing DOACs are unlikely to be ever performed. Indeed, a recent meta-analysis of RW data showed intriguing differences among DOACs in terms of risk of both stroke and major gastrointestinal bleeding, although no differences were seen for intracranial bleeding or all-cause mortality [14]. Although initial experience with DOACs demonstrated consistent advantages over VKA also in terms of adherence and persistence [15], more recent pooled evidence has highlighted that up to 30% of NVAF patients with AF may be nonadherent to DOACs within one year from the initial treatment [16].

To date, two cross-sectional studies compared patients meeting eligibility criteria of DOAC pivotal trials with patients treated in a RW setting in the United Kingdom and Australia [17,18]. These two studies found that patients enrolled within the Randomized Evaluation of Long-Term Anticoagulation Therapy (RE-LY) and Apixaban for Reduction in Stroke and Other Thromboembolic Events in Atrial Fibrillation (ARISTOTLE) trials, respectively, were more representative of the AF population in the RW, in contrast with AF patients enrolled within the Rivaroxaban Once Daily Oral Direct Factor Xa Inhibition Compared with Vitamin K Antagonism for Prevention of Stroke and Embolism Trial in Atrial Fibrillation (ROCKET-AF) study, who were a more narrowly defined group of patients at higher risk of stroke. There have been no Italian studies so far comparing population characteristics and outcomes among DOAC users in the RCT vs. the RW setting. The present large-scale study was aimed at assessing the generalizability of findings of the DOAC pivotal trials by exploring the differences in clinical characteristics and reported efficacy and safety outcome rates between the populations recruited in these studies and populations receiving DOACs for stroke prevention in atrial fibrillation in a RW setting from southern Italy from 2013 to 2017 and comparing drug discontinuation, safety, and effectiveness in these two populations.

## 2. Results

In the pivotal RCTs, 42,336 DOAC users with NVAF were enrolled: 6015 (14.2%) and 6076 (14.3%) were respectively treated with low-dose and high-dose dabigatran [7], 7111 (16.8%) with rivaroxaban [8], 9120 (21.6%) with apixaban [9], and 7002 (16.5%) and 7012 (16.6%), respectively, with low-dose and high-dose edoxaban [10]. In the RW setting, 14,331 incident DOAC users were identified; of these, 7092 (49.5%) were identified as treated because of NVAF: 794 (11.2%) and 635 (8.9%) were respectively treated with low-dose and high-dose dabigatran, 3028 (42.7%) with rivaroxaban, 2113 (29.8%) with apixaban, and 253 (3.6%) and 269 (3.8%), respectively, with low- and high-dose edoxaban (Table 1).

Overall, a similar proportion of males and females were treated with DOACs in the RW setting (50.2% males vs. 49.8% females), compared with the distribution observed in the four RCTs (72.8% males vs. 27.2% females). However, stratification by individual DOAC showed that incident RW users of apixaban, edoxaban, and low-dose dabigatran were more commonly females than males.

Overall, RW patients were older (76.0 ± 8.6 years) than those enrolled in RCTs (70.7 ± 9.2 years) (*p*-value < 0.001). This difference was even more striking for low-dose dabigatran (79.8 ± 7.2 years vs. 71.4 ± 8.6 years) [7] and edoxaban (83.1 ± 7.5 years vs. 70.6 ± 9.3 years) [10] users. In particular, 38.3% of the RW population was ≥ 80 years old (59.0% of RW low-dose dabigatran users).

The mean CHADS_2_ score was comparable between the two settings, except for rivaroxaban users in ROCKET-AF [8] having a statistically significant higher mean CHADS_2_ score compared with the CHADS_2_ score of RW users (3.5 ± 0.9 vs. 2.2 ± 1.4; *p*-value < 0.001). All RW patients had a mean CHA_2_DS_2_ VASC score > 2, higher in the low-dose dabigatran (4.0 ± 1.5) and edoxaban (4.5 ± 1.5) group compared to the other groups. More than half of patients were previously treated with VKAs (56.6% in RCTs vs. 54.3% in the RW).

In general, compared with patients enrolled in other DOAC pivotal RCTs, as well as compared with the RW setting, a higher proportion of patients enrolled in the ROCKET-AF study had a history of comorbidities [8].

Compared with the RCT population, a lower proportion of RW patients had a prior history of stroke/TIA (i.e., 28.5% in RCTs [7,8,9,10] vs. 16.9% in the RW, *p*-value < 0.001), with a more pronounced statistically significant difference among rivaroxaban users: 3916 (55.1%) patients from the ROCKET-AF trial [8] vs. 416 (13.7%) patients in the RW setting. Regarding heart failure, no major differences emerged between the RCT and RW populations, except for rivaroxaban users (62.8% in the ROCKET-AF [8] vs. 35.1% in the RW setting) and high-dose edoxaban users (58.4% in the ENGAGE [10] vs. 40.5% in the RW setting).

A lower proportion of RW DOAC users had a medical history of hypertension than RCT DOAC users (63.6% vs. 87.6%; *p* value < 0.001). Although trials did not report the proportion of subjects suffering from chronic kidney disease (CKD), with the exception of the ARISTOTLE trial [9], 745 (10.5%) RW DOAC users had CKD; for 447 (60.0%) of these, CKD stage was known. Of a total of 305 DOAC users with known moderate–severe CKD, 250 (82%) patients were treated with low-dose DOACs (data not shown).

During a comparable follow-up of approximately two years, the comparison of discontinuation in the RW and RCT settings yielded findings that were not consistent for each DOAC. For example, compared with discontinuers in the RCT setting, a higher proportion of RW dabigatran users (30.4% [7] vs. 19.6%) discontinued their treatment (Figure 1). However, a lower proportion of RW apixaban (15.9% vs. 25.3% [9]) and rivaroxaban (20.4% vs. 23.7% [8]) users discontinued the treatment during the follow-up (*p*-value < 0.001). An exploratory analysis showed that 2.4% of DOAC discontinuers received cardioversion after discontinuation, 21.1% switched to a different DOAC/dosage of DOAC, 36.9% restarted after discontinuation, and 39.6% did not receive any drug after discontinuation (data not shown).

Among incident DOAC users from the RW setting, who were matched to the RCT population, 550 (69.3%), 457 (72.0%), 1405 (46.4%), and 1082 (51.2%) low- and high-dose dabigatran, rivaroxaban, and apixaban users, respectively, had overlapping characteristics (i.e., age, sex, and CHADS_2_ score distribution) with the pivotal RCT populations. As shown in Table 2, less than 3% of RW DOAC users experienced at least one ischemic stroke event during the treatment. It was lower in RW users of high-dose than low-dose dabigatran, both in the RCT (0.93% vs. 1.34% per year) [7], unmatched (0.40% vs. 1.55% per year), and matched (0.11% vs. 1.35% per year) RW settings. Concerning the safety outcomes, as found in RCTs, the rate of gastrointestinal bleeding events was higher than the rate of intracranial bleeding events in the two RW settings (i.e., unmatched and PS matched population). However, in general, gastrointestinal events were less common in the RW than in RCTs, except for apixaban (0.76% vs. 0.79% vs. 0.81% per year in RCT [9], RW matched, and RW unmatched populations, respectively).

## 3. Discussion

Pivotal RCTs have shown the noninferiority/superiority of DOACs to warfarin for ischemic stroke prevention in NVAF and a better safety profile with respect to intracranial bleeding. The present observational study showed how there are major differences between the baseline characteristics of NVAF patients newly treated with DOACs in a RW southern Italian setting compared to patients enrolled in pivotal RCTs. Our results are consistent with several RW studies on DOAC use and outcomes [18,19,20]. In an Australian RW study, clinical characteristics of patients receiving dabigatran and apixaban were fairly more similar with those included in RCT than rivaroxaban [18]. Importantly, the high burden of comorbidities or a very low risk of stroke in the RW population would have not qualified up to one third of RW patients for the inclusion in pivotal RCTs [18]. RW studies, like the present one, complement and expand data about the efficacy and safety of DOACs by specifically including subgroups that have only been marginally represented in RCTs. The main finings in our RW population are as follows.

Firstly, in RCTs, DOAC use was generally more common in males than females compared to a similar sex distribution in the RW. This has important implications about DOAC effectiveness and safety in females, as they frequently use DOACS but are under-represented in RCTs. A recent Italian study identified patients with NVAF who were prescribed a DOAC through the AIFA drug registries during the same study period as the present study (i.e., 2013–2017) [19]; these web-based monitoring registries are one of the many instruments adopted by the Italian National Health Service (NHS) in order to manage budget impact, uncertain clinical outcomes, and appropriate use of DOACs [19]. This study collected data for 683,172 patients from all the Italian regions, except the Emilia Romagna region. The authors showed an overall equal distribution of DOAC use among males and females (50%), fully in line with the distribution seen in our study. However, our results showed that the use of apixaban and low-dose dabigatran and edoxaban was more common among females compared with RCTs. This was also confirmed by the Italian drug registry study, although these results were not stratified by DOAC dosage [19].

In general, women had a different baseline stroke risk than men and different responsiveness to treatments cannot be excluded. According to recent European guidelines [21], female sex is an age-dependent stroke risk modifier rather than a risk factor per se. The simplified CHA2DS2-VASc score could guide the initial decision about OAC treatment in AF patients, but not considering the sex component would underestimate stroke risk in women with AF. In the presence of >1 non-sex stroke risk factor, women with AF consistently have significantly higher stroke risk than men. In general, it is known that more NVAF females than males are treated with DOACs in the RW [18,22].

Overall, RW patients were older than those enrolled in RCTs. This appears to be an information gap concerning elderly persons. The importance of this is highlighted by the age distribution of DOAC users in the RW (38% of RW DOAC users were aged ≥ 80 years old). Indeed, a greater mean age of NVAF patients treated with DOACs in the RW has also been seen in other RW studies [18,19,20]. In particular, Olimpieri et al. found that 19% of DOAC users were aged at least 85 years [19] vs. 15% in our study. According to the dabigatran summary of product characteristics (SPC), a dose reduction is recommended in NVAF patients aged at least 80 years old [23]. However, only 59% of RW low-dose dabigatran users were aged ≥ 80 years old.

The mean CHADS_2_ scores of both the RCT and RW settings were similar, except for ROCKET-AF rivaroxaban users, who had a higher CHADS_2_ score compared with RW rivaroxaban users and other pivotal RCTs enrolled patients. It is important to note that a high-risk patient population was intentionally enrolled in ROCKET-AF (mean CHADS_2_ score = 3.5). Data for patients with a CHADS_2_ score < 2 in ROCKET-AF were not available, and only 13% of this trial population had a CHADS_2_ score < 3 [8].

The worse clinical conditions of patients enrolled in the ROCKET-AF study were confirmed by comparing comorbidities in the RCT vs. RW settings. Compared with the RCT population, a statistically significantly lower proportion of RW patients had a history of hypertension (63.6% vs. 87.6%). The study by Olimpieri et al. using drug registries found a higher proportion of hypertensive patients among DOAC users than the present study (86.5%) [19]. The lower proportion of hypertensive patients in our study may be due to the way this disease was identified, i.e., from hospital discharge diagnoses codes or from codes exempting patients from healthcare service co-payment. This may have led to an underestimation of hypertension as it may not be recorded as a primary or secondary diagnosis code, or as the main disease justifying exemption from co-pay. Moreover, a lower proportion of RW DOAC users had history of stroke/TIA compared with the RCT population. Our results are in line with Olimpieri et al.’s findings on the medical history of stroke in DOAC users based on the entire Italian population: 28.5% in RCTs vs. 16.9% in Palermo RW vs. 18.3% in the drug registry populations [19]. Concerning renal disease, although a direct comparison with the trials was not possible, our study showed that 10.5% of incidental RW DOAC users had a history of CKD. According to the DOAC SPCs, a dose reduction is recommended in NVAF patients with moderate severe CKD (i.e., CrCL < 60 mL/min). As shown in an exploratory analysis on a total of 305 DOAC users with known moderate–severe CKD, 82% were treated with low-dosage DOACs. It is unclear why the remaining 18% of patients were treated with high-dosage DOACs. Yao et al. showed that 43.0% of 1473 patients with renal disease who should have had DOAC dose reduction actually were overdosed [24]. Concerning edoxaban, a dose reduction is recommended in patients with ≥1 of the following clinical factors: moderate–severe renal impairment, low body weight (≤60 kg), or concomitant use of P-glycoprotein inhibitors. No dose reduction is required for the elderly. Our results showed that only one third of low-dose edoxaban users in a RW setting had history of CKD and among patients with known CKD stage and no renal indication for dose reduction (i.e., CrCl > 50 mL/min), almost 12% were underdosed, in line with Yao et al.’s study (e.g., 13%) [14]. Data on body weight were not available from the Palermo LHU database and none of the incidental edoxaban users were treated with P-glycoprotein inhibitors. In general, the estimates of NVAF patients with a history of comorbidities in our RW population were similar to estimates reported in the Italian RW population, although Olimpieri et al. did not stratify by DOAC dosage [19], and by the latest National Report on Medicines Use in Italy [25].

The present study provides some interesting findings on discontinuation in the RW setting compared to the RCT setting. As expected, discontinuation rates were different in the RW setting. Among RW DOAC initiators, the proportion of patients discontinuing treatment over a median follow-up of almost two years was 30% (versus 20% in RE-LY) for dabigatran, 20% (versus 24% in ROCKET AF) for rivaroxaban, and 16% (versus 25% in ARISTOTLE) for apixaban. Our results are in line with findings from recent observational studies [26,27,28]. In particular, a long-term study on the use of DOACs in clinical practice conducted by the Karolinska Institute showed that discontinuation rate was lower for apixaban (11.5, 7.5–16.8) users compared to rivaroxaban (23.9, 18.6–30.1, *p*-value = 0.001) and dabigatran (30, 23.4–37.9, *p*-value < 0.001 30%) users [26], in line with our results. Other RW studies showed a higher rate of discontinuation than those seen in our study, but with the same trend among individual DOAC groups. A population-based cohort study using a UK primary care setting found that the discontinuation rates were respectively 40.0% for dabigatran, 29.6% for rivaroxaban, and 26.1% for apixaban during the first year of treatment [27]. A recent meta-analysis on RW data comparing persistence of DOAC users treated for NVAF showed that the pooled proportion of persistence for all follow-up durations (3–36 months) from 36 studies was 69% [28]. Subgroup analyses by individual DOACs showed a pooled discontinuation of 38% for dabigatran, 28% for rivaroxaban, and 26% for apixaban. The reason for discontinuation was not a focus of this retrospective study; however, there is evidence to suggest that the high rate of discontinuation could be due to side effects of DOACs, (e.g., dyspepsia, tiredness, itching, etc.) and mostly bleeding events [26,29]. However, DOAC nonpersistence is associated with an increased risk of stroke/TIA [30]. Moreover, the discontinuation of DOACs could be justified if sinus rhythm has been maintained for 1–3 months after a cardioversion, especially in low-risk AF patients younger than 65 years old with no cardiovascular comorbidities. An exploratory analysis showed that 2.4% of DOAC discontinuers received cardioversion after discontinuation, 21.1% switched to a different DOAC/dosage of DOAC, 36.9% restarted after discontinuation, and 39.6% did not receive any drug after discontinuation.

Our results showed that the efficacy and safety event rates in the RCT population were slightly higher compared to the matched RW population, but were generally comparable. These differences were more marked for major bleeding with high-dose dabigatran, while in the case of apixaban they were less notable. In general, the effectiveness and safety profile of DOACs have been confirmed by RW data [31,32], through a comparison with warfarin or between the three DOACs. Specifically, data from observational studies showed a favorable bleeding risk profile for dabigatran and apixaban compared with rivaroxaban [32], as shown by our results. Concerning the specific type of bleeding, our results showed that gastrointestinal bleeding rate was higher than intracranial bleeding rate in the RW setting, in line with RCT results. However, rates of gastrointestinal bleeding were lower in the RW setting than those observed in the RCT setting, especially in high-dose dabigatran and rivaroxaban users. Certainly, the high value of the mean CHADS_2_ score of the rivaroxaban trial population, reflecting the serious health conditions of the subjects enrolled, may justify the high rates reported. Specifically, for rivaroxaban, the ROCKET-AF study showed that it was equivalent to warfarin for risks of major bleeding and mortality, while there was a significant reduction in intracranial bleeding risk compared with warfarin. In RW, studies based on Danish national registries [33,34] have reported a higher risk of mortality in rivaroxaban users against warfarin users, but these findings have not been supported by other studies.

Our study must be interpreted in light of its limitations. First, although the claims database used provides a broad range of clinical information, some clinical aspects that were used in the inclusion or exclusion criteria were not available. Second, the assessment of bleeding risk (e.g., HAS-BLED) could not have been evaluated because the international normalized ratio (INR) was not available in the RW setting. Third, NVAF as a diagnosis was probably underestimated among DOAC users, because neither the Italian Drug Agency drug registries, nor the electronic therapeutic plans related to NVAF from the Palermo claims database were available. However, a manual inspection of 4% of paper-based therapeutic plans was carried out to validate the NVAF diagnoses identified from the claims database. Third, laboratory data were not available; for this reason CKD stage identification, through ICD 9-CM codes, could be underestimated. Moreover, an underestimation of stroke and major bleeding event rates could be possible because diagnoses for these conditions were only identified if they were recorded as a primary/secondary diagnosis in the Palermo LHU database—all hospitalizations of study subjects enrolled in the Palermo LHU database and those that occurred in hospitals different from hospitals affiliated to Palermo LHU were not traced. On the other hand, it is important to say that the Palermo databases are of high quality, informative, and representative of the general patient population and contain a multitude of information that allowed us to follow the patients’ stories for a long time, so as to have comparable follow-up to those of the trials of interest, of approximately two years.

## 4. Materials and Methods

### 4.1. Data Source

The four pivotal phase III RCTs (RE-LY, ARISTOTLE, ROCKET-AF, and ENGAGE AF-TIMI 48) comparing the efficacy of dabigatran, apixaban, rivaroxaban, and edoxaban to warfarin in terms of primary efficacy and safety outcomes in NVAF patients were considered. The pivotal trials registered in clinicaltrials.gov [35], as well as corresponding original articles, protocols, and appendices, were used as data sources to identify population characteristics and clinical outcomes of treatment with DOAC use in an experimental setting, while the claims database of the Palermo Local Health Unit (LHU) from southern Italy was used to identify these in a RW setting.

Fully anonymized data were extracted from the Palermo LHU claims database, covering a total population of 1,362,708 persons from 2013 to 2017. Individual patient-level data on dispensed drugs that are covered by the Italian NHS, hospital discharge forms, emergency department visits, and exemptions from co-payment for the healthcare service diagnostic tests were retrieved. Coverage of all these healthcare services is very high since Italy has a NHS offering universal care for all residents in each region. Drugs were coded using the Anatomical Therapeutic Chemical (ATC) classification system and the Italian marketing authorization code (AIC), while indication of use and causes of hospitalization were coded using the International Classification of Disease, 9th revision, Clinical Modification (ICD 9-CM). All the population characteristics of pivotal RCTs and Palermo claims database are reported in Appendix A.

### 4.2. Study Population

In the pivotal RCTs, patients with a diagnosis of NVAF based on electrocardiogram results were included. From the Palermo claims database, all incident DOAC users (no DOAC dispensing within one year prior to the first DOAC dispensing date, i.e., index date—ID) with at least 1 year of database history and treated for NVAF were identified and grouped by molecule (i.e., dabigatran, edoxaban, apixaban, and rivaroxaban). Each DOAC group was further classified in line with the dosing regimen investigated in pivotal RCTs: low- or high-dose dabigatran and edoxaban, and any dose of apixaban/rivaroxaban. The indication of use (i.e., NVAF was identified from hospital discharge diagnoses [ICD 9-CM codes: 427.3× (diagnosis) or 99.61 (procedure)] was evaluated any time prior to ID.

### 4.3. Study Drugs

All the marketed DOACs were included in the study: dabigatran, rivaroxaban, apixaban, and edoxaban. In line with RE-LY and ENGAGE AF-TIMI 48 studies, incidental users of dabigatran (110 mg vs. 150 mg, twice a day) and edoxaban (30 mg vs. 60 mg, once a day) were categorized as low-dose or high-dose users at ID. Since no distinction by dosage was made in the pivotal trials of rivaroxaban and apixaban, no dosage-specific groups (standard dose and reduced dose) for these drugs were considered (Appendix A).

### 4.4. Data Analysis

To compare the pivotal RCT vs. RW population, the following analyses were carried out:

#### 4.4.1. Description of Demographic and Clinical Characteristics of DOAC Users

Baseline characteristics of NVAF patients who were newly treated with each specific DOAC in the RW setting were compared with those of the respective pivotal RCT population. At baseline, the following variables were assessed and compared in the two settings: patient demographic characteristics (gender and age) evaluated at ID, CHADS_2_ score and CHA_2_DS_2_ VASC score, comorbidities (e.g., stroke/TIA, heart failure, diabetes, chronic kidney disease, and hypertension), and vitamin K antagonist use evaluated any time prior to ID.

Concerning age, very old (≥80 years) RW DOAC users were identified and analyzed separately. Pivotal RCTs did not report maximum age-related inclusion criteria.

CHADS_2_ score includes the following stroke risk factors: age ≥ 75 years, history of chronic heart failure, hypertension, and/or diabetes (assigned 1 point each), or history of stroke, or TIA (assigned two points each). A score of 1 indicates an intermediate risk of stroke, a score of ≥2 indicates a high risk of stroke. CHA_2_DS_2_ VASC score includes the following stroke risk factors: age (65–74 years), sex (female gender), history of congestive heart failure, hypertension, vascular disease, and/or diabetes (assigned 1 point each), or age (≥75 years), history of stroke/TIA/thromboembolism (assigned two points). A score of ≥2 indicates a high risk of stroke. Concerning comorbidities, the ARISTOTLE study protocol provided information on the renal function of enrolled patients; on the other hand, the RE-LY, ENGAGE AF-TIMI 48, and ROCKET-AF study protocols reported that subjects with severe renal impairment (CrCL ≤ 30 mL/min) were excluded. For the ENGAGE AF-TIMI 48 study specifically, the drug dosage was halved (i.e., 30 mg edoxaban rather than 60 mg) for study patients with moderate renal impairment (30 mL/min ≤ CrCL ≤ 50 mL/min). An exploratory analysis identified RW users with history of CKD, using ICD 9-CM codes from hospitalizations/emergency department visits, or healthcare service co-payment exemptions code. Of these, all patients with known CKD stages (through ICD 9-CM codes from hospitalizations/emergency department visits) were identified and the proportion of patients with moderate–severe renal impairment (e.g., CKD stages III–V and dialysis) was calculated.

#### 4.4.2. DOAC Treatment Discontinuation

In pivotal RCTs, discontinuation of DOAC treatment was evaluated as interruption of the therapy due to patient decision or outcome/serious adverse event occurrence. In the RE-LY study, discontinuation was evaluated within the first and the second year of treatment, whereas in the other pivotal RCTs it was evaluated during the entire patient follow-up (median follow-up: 1.8–1.9 years; interquartile range [IQR] not reported). Based on the number of dispensed tablets and dosage regimens, DOAC treatment discontinuation in the RW population was estimated among incident users, and compared with that from the respective pivotal trials at the same time points.

Specifically, an incident DOAC user was considered as a discontinuer in the presence of a treatment gap exceeding 90 days between refill for the same DOAC. The follow-up of each incident DOAC user (provided they were still on therapy) was censored at the end of the study period (31 December 2017), or date of death, whichever came first. Edoxaban users were excluded from this analysis because this drug was approved in Italy in February 2016, and reached the market in Palermo LHU in November 2016, as a result of which the median follow-up for edoxaban users was much shorter than that of the ENGAGE trial (0.5 vs. 2.8 years).

#### 4.4.3. Clinical Outcomes

In the RW population, the event rates of ischemic stroke (as a measure of effectiveness) and intracranial and gastrointestinal bleedings (as a measure of safety) during each DOAC use were estimated and compared with event rates of the same efficacy and safety outcomes as reported in the DOAC pivotal trials. In the RW setting, clinical outcomes were identified through search of specific ICD9-CM codes from hospital admissions and emergency department visits (Appendix A). Only clinical outcomes occurring between the first DOAC dispensing and the date of discontinuation, therapeutic switching, date of death, or end of follow-up were considered. Outcome occurrence was stratified by DOAC type, excluding edoxaban because of the short follow-up of edoxaban users. If patients experienced the same study outcome more than once, the first outcome was considered. All the estimates of clinical outcomes were also calculated in a simulated RCT population, by matching by age, sex, and CHADS_2_ score, individual incident RW DOAC users and RTCs DOAC users.

### 4.5. Statistical Analysis

To assess and compare demographic and clinical characteristics of DOAC users in pivotal RCTs vs. the RW setting, descriptive statistics were performed. Results were reported as mean ± standard deviation (SD) for continuous variables, and as absolute frequencies and percentages for categorical variables. The event rate of each specific clinical outcome was calculated as the number of events occurring during DOAC use divided by the number of 100 person-years, for each DOAC user separately in the RW and RCT settings. In order to evaluate whether the outcome estimates reported in the RCTs were reproducible using data from the RW population sample, each patient from the RW population was matched with a patient from the simulated RCT population. Specifically, for each trial and each DOAC treatment arm, patient-level RCT pseudo-data were simulated (on the basis of their available summary statistics) 1000 times using a Monte-Carlo-based approach and, at each iteration, this simulated dataset was 1:1 matched with the population dataset on the basis of their propensity score, using Parsons’ greedy 5 to 1 digit match algorithm [36]. Propensity score methods were already used to assess the external validity of RCTs [37]. This is the individual probability of being selected in the simulated RCT dataset and was estimated by a multivariable logistic regression model, which included subjects’ age, sex, presence of stroke/TIA, heart failure, diabetes, hypertension, and CHADS_2_ score as covariates. At each iteration, both efficacy and safety outcomes were assessed on the matched population sample and the event rates were computed. At the end of all iterations, the medians of the empirical distributions of all outcome event rates were calculated across all simulations (Appendix A). A two sided *p*-value < 0.05 was considered for statistical significance. All statistical analyses were performed using SAS Release 9.4.

## 5. Conclusions

Characteristics of NVAF patients treated with DOAC in a RW setting are substantially different from patients enrolled in pivotal trials (e.g., much higher proportion of very old patients and females). Except for dabigatran, a lower proportion of DOAC discontinuers was observed in the real world than in a pivotal trial setting. Compared with the RCT setting, ischemic stroke rate appeared to be similar in the RW setting, except for high-dose dabigatran, while bleeding outcomes appeared to be less common in the RW. This study provides reassurance to practicing physicians that DOAC use appears effective in stroke prevention and likely safer in RW population than in RCT population. These results may be related to a lower burden of comorbidities, despite more advanced age in the RW population compared to pivotal RCTs population. Observational studies using healthcare databases are important tools to rapidly explore the generalizability of findings from pivotal RCTs to a real-world setting.

## Figures and Tables

**Figure 1 pharmaceuticals-14-00290-f001:**
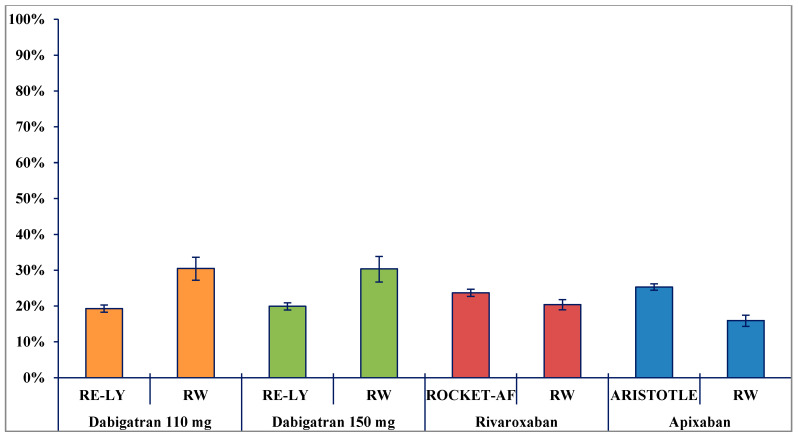
Distribution of direct oral anticoagulant (DOAC) discontinuers in pivotal RCT populations and in Palermo RW populations during the follow-up. Note: all differences were statistically significant (*p*-value < 0.001). Median (interquartile range–IQR) follow-up—years: Dabigatran 110 mg: Randomized Evaluation of Long-Term Anticoagulation Therapy (RE-LY) = 2.0 (IQR not reported); Palermo RW population = 1.8 (0.8–2.7). Dabigatran 150 mg: RE-LY = 2.0 (IQR not reported); Palermo RW population = 2.0 (0.9–3.0). Rivaroxaban: ROCKET-AF = 1.9 (IQR not reported); Palermo RW population = 1.7 (0.8–2.6). Apixaban: ARISTOTLE = 1.8 (IQR not reported); Palermo RW population = 1.4 (0.7–2.2).

**Table 1 pharmaceuticals-14-00290-t001:** Comparison of baseline characteristics of real-world (RW) population vs. randomized controlled trial (RCT) population, stratified by individual drug and dosage.

	DABIGATRAN 110 mg		DABIGATRAN 150 mg		RIVAROXABAN		APIXABAN		EDOXABAN 30 mg		EDOXABAN 60 mg	
	RE-LY *n* = 6015	RW *n* = 794	RE-LY *n* = 6076	RW *n* = 635	ROCKET- AF *n* = 7111	RW *n* = 3028	ARISTOTLE *n* = 9120	RW *n* = 2113	ENGAGE AF-TIMI 48 *n* = 7002	RW *n* = 253	ENGAGE AF-TIMI 48 *n* = 7012	RW *n* = 269
Sex—*n* (%)												
Males	3868 (64.3)	382 (48.1)	3840 (63.2)	347 (54.6)	4292 (60.4)	1553 (51.3)	5886 (64.5)	1031 (48.8)	4285 (61.2)	92 (36.4)	4354 (62.1)	153 (56.9)
Females	2147 (35.7)	412 (51.9)	2236 (36.8)	288 (45.4)	2819 (39.6)	1475 (48.7)	3234 (35.5)	1082 (51.2)	2717 (38.8)	161 (63.6)	2658 (37.9)	116 (43.1)
Age, y—mean (±SD)	71.4 (±8.6)	79.8 (±7.2)	71.5 (±8.8)	69.2 (±8.5)	71.2 (±9.4)	75.0 (±9.7)	69.1 (±9.6)	76.1 (±9.6)	70.6 (±9.3)	83.1 (±7.5)	70.6 (±9.5)	72.9 (±9.3)
CHADS_2_ score—mean (±SD)	2.1 ± 1.1	2.5 ± 1.3	2.1 ± 1.2	1.7 ± 1.3	3.5 ± 0.9	2.2 ± 1.4	2.1 ± 1.1	2.4 ± 1.4	2.8 ± 1.0	2.9 ± 1.4	2.8 ± 1.0	2.1 ± 1.3
CHA_2_DS_2_ VASc score—mean (±SD)	n.a.	4.0 ± 1.5	n.a.	2.9 ± 1.6	n.a.	3.6 ± 1.7	n.a.	3.8 ± 1.7	n.a.	4.5 ± 1.5	n.a.	3.4 ± 1.6
Previous vitamin K antagonist use ^a^—*n* (%)	3011 (50.1)	453 (57.1)	3049 (50.2)	377 (59.4)	4430 (62.3)	1747 (57.7)	5208 (57.1)	1035 (49.0)	4145 (59.2)	108 (42.7)	4123 (58.8)	132 (49.1)
Comorbidities ^a^—*n* (%)												
Stroke/TIA	1195 (19.9)	138 (17.4)	1233 (20.3)	72 (11.3)	3916 (55.1)	416 (13.7)	1748 (19.2)	491 (23.2)	2006 (28.6)	50 (19.8)	1976 (28.2)	35 (13.0)
Heart failure	1937 (32.2)	300 (37.8)	1934 (31.8)	180 (28.3)	4467 (62.8)	1063 (35.1)	3235 (35.5)	828 (39.2)	3979 (56.8)	148 (58.5)	4097 (58.4)	109 (40.5)
Diabetes mellitus	1409 (23.4)	266 (33.5)	1402 (23.1)	190 (29.9)	2878 (40.5)	1020 (33.7)	2284 (25.0)	737 (34.9)	2544 (36.3)	93 (36.8)	2559 (36.5)	87 (32.3)
Hypertension	4738 (78.8)	513 (64.6)	4795 (78.9)	383 (60.3)	6436 (90.5)	1917 (63.3)	7962 (87.3)	1374 (65.0)	6575 (93.9)	165 (65.2)	6591 (94.0)	161 (59.8)
CKD	n.a.	76 (9.6)	n.a.	21 (3.3)	n.a.	272 (9.0)	5319 (58.3)	278 (13.1)	n.a	82 (32.4)	n.a.	16 (5.9)

Legend: SD = standard deviation; TIA = transient ischemic attack; CKD = chronic kidney disease; n.a. = not available. ^a^ Evaluated any time prior to ID.

**Table 2 pharmaceuticals-14-00290-t002:** Efficacy and safety outcome rates in RCT population vs. RW unmatched population vs. RW 1:1 matched population.

	DABIGATRAN 110 mg			DABIGATRAN 150 mg			RIVAROXABAN			APIXABAN		
	RE-LY *n* = 6015	Unmatched RW *n* = 794	Matched RW *n* * = 550	RE-LY *n* = 6076	Unmatched RW *n* = 635	Matched RW *n* * = 457	ROCKET- AF *n* = 7061	Unmatched RW *n* = 3028	Matched RW *n* * = 1405	ARISTOTLE *n* = 9120	Unmatched RW *n* = 2113	Matched RW *n* * = 1082
**Efficacy- *n* (events per 100 person-years)**												
Ischemic stroke	1.34	1.55	1.35	0.93	0.40	0.11	1.34	1.42	1.10	0.97	1.33	0.96
**Safety- *n* (events per 100 person-years)**												
Major bleeding	2.92	2.20	2.28	3.40	0.97	1.02	3.60	2.44	2.31	2.13	2.00	1.94
Intracranial bleeding	0.23	0.14	0.10	0.32	0.24	0.23	0.49	0.46	0.28	0.33	0.23	0.18
Gastrointestinal bleeding	1.13	1.05	1.08	1.60	0.32	0.34	2.00	0.92	0.99	0.76	0.81	0.79

Legend: RCT: randomized controlled trials; RW: real world; * median number of 1:1 matched subjects detected in Palermo database calculated across 1000 Monte Carlo simulations.

## Data Availability

Fully anonymized dataset is available only upon request to the corresponding author as there is an agreement between University of Messina and data provider (Palermo Local Health Unit) not to share the data publicly.

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
