# Peer review of "Comparison of Direct Oral Anticoagulant Use for the Treatment of Non-Valvular Atrial Fibrillation in Pivotal Clinical Trials vs. the Real-World Setting: A Population-Based Study from Southern Italy"

_pharmaceuticals, 2021, doi:10.3390/ph14040290_

Round 1

Reviewer 1 Report

In this manuscript, Ingrasciotta and colleagues performed an analysis on an administrative database on the comparison between Real-World vs. RCTs-enrolled patients treated with DOACs for atrial fibrillation. The aim was to evaluate potential differences that may impact the generalizability of the findings of RCTs in clinical practice.
This topic, however, is not new. A lot of studies already addressed the topic, reporting data from real-world cohorts of AF patients treated with OAC; efficacy and safety of DOACs in RW, as well as risk of discontinuation, have already been addressed both in large cohorts studies (e.g. see the GLORIA-AF study for Dabigatran) and in landmark meta-analysis, so that the novelty and added value of this article is very limited, although the authors should be commended for their efforts in providing a robust methodology for their study.
Other than this major concern, there are several issues that further limits this study.

My comments in detail:
- The introduction should be expanded to cite and reflect existing literature on real-world use of DOACs in atrial fibrillation, both from the point of view of prevention of thromboembolism and bleeding events, and discontinuation/persistence.
- In the introduction, the author state that the study "was aimed at evaluating the generalisability [...] by comparing the populations recruited in these studies to populations of NVAF patients meeting the same eligibility criteria but identified from a RW setting". In my opinion, this is not useful: as the authors clearly stated in the introduction, patients from the RW setting may differ broadly from those recruited in RCTs, so clinical importance of establishing a RW-based cohort similar to that enrolled in RCTs is very limited and add low information to clinical practice.
- Results are currently presenting mainly with figures; however, to improve clarity, I feel that a table with the comparison between landmark RCTs and the RW population may be more useful and clear for the reader, to understand the differences between RW and RCTs cohorts. Moreover, baseline characteristics of the RW cohort are presented only according to the comparison to the RCTs; however, in my opinion, it would be better to report a more comprehensive set of variables, including the updated thromboembolic risk stratification score (CHA2DS2-VASc), assessment of bleeding risk (HAS-BLED) etc. etc.
- Were these patients naive for the use of OAC (i.e. VKA) or were they shifted from VKA to DOACs? This is very important since this may have affected rates of clinical events (possibly being different among naive users vs. "shifters").
- In the definition of discontinuation, is unclear whether a patient was count as "discontinuating" even if transitioned to another DOAC. This is a non-rare occurrence in clinical practice, so that it may led to overeestimation of Discontinuation for these patients.

Author Response

Response to Reviewer 1 Comments

Reviewer 1:

Comments and Suggestions for Authors

In this manuscript, Ingrasciotta and colleagues performed an analysis on an administrative database on the comparison between Real-World vs. RCTs-enrolled patients treated with DOACs for atrial fibrillation. The aim was to evaluate potential differences that may impact the generalizability of the findings of RCTs in clinical practice.

This topic, however, is not new. A lot of studies already addressed the topic, reporting data from real-world cohorts of AF patients treated with OAC; efficacy and safety of DOACs in RW, as well as risk of discontinuation, have already been addressed both in large cohorts studies (e.g. see the GLORIA-AF study for Dabigatran) and in landmark meta-analysis, so that the novelty and added value of this article is very limited, although the authors should be commended for their efforts in providing a robust methodology for their study.

Other than this major concern, there are several issues that further limits this study.

Response: to our knowledge this is the first study specifically evaluating generalizability of DOAC RCT in Italian real-world setting. We adopted an innovative approach where each patient from the RW population was matched with a patient from the simulated RCT population in order to evaluate whether the outcome estimates reported in the RCTs were reproducible using data from the RW population sample.

My comments in detail:

  1. The introduction should be expanded to cite and reflect existing literature on real-world use of DOACs in atrial fibrillation, both from the point of view of prevention of thromboembolism and bleeding events, and discontinuation/persistence.

Response 1: We thank the reviewer for this valuable comment. We have expanded the Introduction section accordingly and added pertinent references.

  1. In the introduction, the author state that the study "was aimed at evaluating the generalisability [...] by comparing the populations recruited in these studies to populations of NVAF patients meeting the same eligibility criteria but identified from a RW setting". In my opinion, this is not useful: as the authors clearly stated in the introduction, patients from the RW setting may differ broadly from those recruited in RCTs, so clinical importance of establishing a RW-based cohort similar to that enrolled in RCTs is very limited and add low information to clinical practice.

Response 2: We thank the reviewer for bringing up this point; we rephrased the aim of the study in a more appropriate way. We hypothesized -and verified seeing by the results of the study- that RW DOAC users and RCT DOAC users differ broadly from each other. We underscored differences between RCT and RW populations and explored if those differences may also account for differences in terms of effectiveness and safety outcomes event rates. The text has been modified accordingly.

  1. Results are currently presenting mainly with figures; however, to improve clarity, I feel that a table with the comparison between landmark RCTs and the RW population may be more useful and clear for the reader, to understand the differences between RW and RCTs cohorts. Moreover, baseline characteristics of the RW cohort are presented only according to the comparison to the RCTs; however, in my opinion, it would be better to report a more comprehensive set of variables, including the updated thromboembolic risk stratification score (CHA2DS2-VASc), assessment of bleeding risk (HAS-BLED) etc. etc.

Response 3: We thank the reviewer for his suggestion; we have accordingly removed figure 1a-d and reported the baseline characteristics of RCTs and RW populations, stratified by individual drug and dosage in the new table 1. Although a comparison with RCT populations was not possible, we have added some other variables, such as the mean CHA2DS2-VASc score (±SD), previous vitamin K antagonist use and, among comorbidities, the history of chronic kidney disease. We have updated the methods and the results section. Unfortunately, the HAS-BLED could not have been evaluated because the international normalized ratio (INR) was not available.

  1. Were these patients naive for the use of OAC (i.e. VKA) or were they shifted from VKA to DOACs? This is very important since this may have affected rates of clinical events (possibly being different among naive users vs. "shifters").

Response 4: As described in the Study population paragraph of the Materials and Methods section, we identified all incident DOAC users (no DOAC dispensing within one year prior to the first DOAC dispensing date, i.e. Index Date). RW patients, as well as RCTs patients, could be previously treated with vitamin K antagonists. As previously mentioned, we have added “previous use of vitamin K antagonist” as a variable in Table 1 and we found that more than half of patients were previously treated with vitamin K antagonists both in RCT and real-world setting (56.6% in RCTs vs. 54.3% in RW).

  1. In the definition of discontinuation, is unclear whether a patient was count as "discontinuating" even if transitioned to another DOAC. This is a non-rare occurrence in clinical practice, so that it may led to over-estimation of Discontinuation for these patients.

Response 5: We thank the reviewer for bringing up this point. As described in the paragraph “4.4.2. DOAC treatment discontinuation” of the Materials and Methods section, an incident DOAC user (in the RW setting) was considered as a discontinuer in presence of a treatment gap exceeding 90 days between refill for the same DOAC of the same strength (i.e. we considered as unit of evaluation the dose-specific DOAC). Based on reviewer’s suggestion, as regards the DOAC users who discontinued the treatment, we explored specifically if they switched to another DOAC or strength of the same DOAC, restarted the same DOAC or discontinued any DOAC therapy in the period following the discontinuation. We found that 39.6% of discontinuers did not receive any DOAC after discontinuation, followed by 36.9% restarting the same DOAC after discontinuation, 21.1% switching to a different DOAC or reducing/increasing the DOAC dosage. In addition, the discontinuation to DOACs could be justified if sinus rhythm had been maintained for 1-3 months after a cardioversion; for this reason, as requested during the first revision of the manuscript, we calculated the proportion of discontinuers receiving cardioversion after discontinuation and we found that 2.4% of DOAC discontinuers received it after discontinuation. We have described this exploratory analysis in the discussion section.

Reviewer 2 Report

Ingrasciotta et al present a comparison of random controlled trials (RCT) to real world (RW) database information obtained from Italian patients concerning the population characteristics and clinical outcomes of patients with non-valvular atrial fibrillation treated with direct oral anticoagulants (DOACs).

The manuscript presented to me appears to be a revision although it is not labeled as such and I have no idea what the previous reviewer criticisms were.

Compared to various international RCT, not surprisingly, the Italian patients had a variety of differences in overall age, male:female ratio, efficacy and safety.  The rates of discontinuance were different as well between the various DOAC studied.

The abstract notes these differences, but does not provide any concluding statement.  There were differences – why should this be important to the readership?

The numbering of figures is strange.  Why have four free-standing figures labeled 1a-d and then have a figure 2?  I suspect that there was an original four panel figure 1.  Just make these figures 1-5.

As with the abstract, the discussion provides no statement to bring the importance of the findings together into a meaningful message.  Why is RW vs RCT comparisons important?  What should we do as practitioners armed with this new information?

Author Response

Response to Reviewer 2 Comments

Reviewer 2:

Comments and Suggestions for Authors

Ingrasciotta et al present a comparison of random controlled trials (RCT) to real world (RW) database information obtained from Italian patients concerning the population characteristics and clinical outcomes of patients with non-valvular atrial fibrillation treated with direct oral anticoagulants (DOACs).

  1. The manuscript presented to me appears to be a revision although it is not labeled as such and I have no idea what the previous reviewer criticisms were.

Response 1: We confirm that this is a revision of a previous submission; as far as we understand, the re-submitted revised version of the manuscript is considered as a new submission. For this reason, the reviewer can see the manuscript as a track-change version.

  1. Compared to various international RCT, not surprisingly, the Italian patients had a variety of differences in overall age, male:female ratio, efficacy and safety. The rates of discontinuance were different as well between the various DOAC studied. The abstract notes these differences, but does not provide any concluding statement. There were differences – why should this be important to the readership?

Response 2: We thank the reviewer for bringing up this point; as also suggested by the other two reviewers, we concluded the abstract specifying that “In conclusion, except for dabigatran, a lower proportion of DOAC discontinuers was observed in real world than in pivotal RCT settings. This study provides reassurance to practicing physicians that DOAC use appears effective in stroke prevention and likely safer in RW patients than in RCT enrolled patients. These results may be related to a lower burden of comorbidities despite more advanced age in the RW population compared to pivotal RCT population”. We have added these considerations in the abstract and in the conclusions sections.

  1. The numbering of figures is strange.  Why have four free-standing figures labeled 1a-d and then have a figure 2?  I suspect that there was an original four panel figure 1.  Just make these figures 1-5.

Response 3: all the tables and figures numbering have been carefully revised and modified as needed. In addition, as suggested by the first reviewer, we removed figure 1a-d and reported the baseline characteristics of RCTs and RW populations in a new table 1.

  1. As with the abstract, the discussion provides no statement to bring the importance of the findings together into a meaningful message.  Why is RW vs RCT comparisons important?  What should we do as practitioners armed with this new information?

Response 4: We thank the reviewer for this valuable comment. We believe that the results of this study may have important implications for patient management, especially for those subgroups who are underrepresented in RCT. We have added relevant comments about these points at the beginning of the discussion, where we underline the additional role of RWE to complement data from RCT and also added some considerations in the conclusions.

Reviewer 3 Report

Here they are my comments:

Abstract

This section needs a conclusion based on the findings.

Introduction

Line 9, the sentence needs a citation to be supported.

Lines 10-17 again needs citation(s) to be supported.

Line 37, also needs a citation to support the sentence.

Line 30, please discuss more and describe in what way, the RCT suffers from limited external validity in general.

Lines 44-45, please provide the full name for abbreviations RE-LY and ARISTOTLE and ROCKET-AF.

Again line 45-52, you have mantioned previous RCTs without any citation to them to help what you are talking about in terms of comparisons. A table in here would help to present these RCTs in terms of author, country, year, sample and setting, data collection and data analysis and conclusion. Please also introduce the Iralian RW study in here to find about it before reading the comparison's results.

Methods

There is a need to provide details of what you have done for comparing the results of RCTs and RW study. Have you performed a meta-analysis or something like this for comparison? Please be elaborative in terms of the design of your research, data collection, data analysis process so on in here and add as much as you can to make your research repeatable by other researchers.

Results

Please do not forget to use citations for any details comes through this section to find which finding belongs to which RCT or RW study.

Discussion

Line 42, please elaborate on it and describe those other RW studies. I suggest to compare your findings with those of RWs studies and systematic reviews in other settings. Therefore, do not reduce the level of your study thorough being compared with single RCTs.

Line 46, which recent Iralian study?

Line 76 needs a citation, also.

Line 122 should be connected to line 121.

Line 122, please describe those observational studies.

I can not understand why Methods has been placed after the Discussion! A figure can help with summarising the research method. The description of the RW study should be separated from the RCTs.

Conclusion

It should contain implications of the study based on the findings. What would be the implications for education, management, policymaking etc?

Author Response

Response to Reviewer 3 Comments

Reviewer 3:

Comments and Suggestions for Authors

Abstract:

  1. This section needs a conclusion based on the findings.

Response 1: Using the suggestions of the other two reviewers, we have added the following sentences: “In conclusion, except for dabigatran, a lower proportion of DOAC discontinuers was observed in real world than in pivotal RCT settings. This study provides reassurance to practicing physicians that DOAC use appears effective in stroke prevention and likely safer in RW patients than in RCT enrolled patients. These results may be related to a lower burden of comorbidities despite more advanced age in the RW population compared to pivotal RCT population”.

Introduction:

  1. Line 9, the sentence needs a citation to be supported.

Response 2: We added the following reference “Zoni-Berisso M, Lercari F, Carazza T, Domenicucci S. Epidemiology of atrial fibrillation: European perspective. Clin Epidemiol. 2014; 213-220”. We moved this reference number at the end of the following sentence “In Italy, in the last decade, the prevalence of AF in the general adult population was 1.9%, increasing with age (0.1%-0.2% in persons aged <49 years versus 10%-17% in persons aged ≥80 years old) [2]”.

  1. Lines 10-17 again needs citation(s) to be supported.

Response 3: We have added the requested references accordingly.

  1. Line 37, also needs a citation to support the sentence.

Response 4: We have added the following reference “Desmaele S, Steurbaut S, Cornu P, Brouns R, Dupont AG. Clinical trials with direct oral anticoagulants for stroke prevention in atrial fibrillation: how representative are they for real life patients? Eur J Clin Pharmacol. 2016 Sep;72(9):1125-34. doi: 10.1007/s00228-016-2078-1. Epub 2016 Jun 7. PMID: 27272167”.

  1. Line 30, please discuss more and describe in what way, the RCT suffers from limited external validity in general.

Response 5: We have described more in details the differences between RCTs and RW studies, as following: “Although randomized controlled trials (RCTs) are considered the ‘gold standard’ for generating evidence, especially on drug efficacy, these studies generally suffer from limited external validity [11], because patients at high risk of adverse effects, with multiple comorbidity or frail elderly persons are often excluded, suggesting the need for the reevaluation of the benefit-risk profile of the drugs in real-world setting. Concerning the DOACs pivotal RCTs specifically, these had a duration ranging from 13 to 54 months. However, it is known that DOACs are used for much longer periods in clinical practice...”.

  1. Lines 44-45, please provide the full name for abbreviations RE-LY and ARISTOTLE and ROCKET-AF.

Response 6: We have spelled-out the names of the pivotal RCTs accordingly.

  1. Again line 45-52, you have mantioned previous RCTs without any citation to them to help what you are talking about in terms of comparisons. A table in here would help to present these RCTs in terms of author, country, year, sample and setting, data collection and data analysis and conclusion. Please also introduce the Iralian RW study in here to find about it before reading the comparison's results.

Response 7: The reviewer probably did not see the Table S1, describing the comparative characteristics (e.g. study design, study years, study population, study drugs - specifying the dose-specific comparison if feasible) of DOACs pivotal trials and Palermo claims database.

Methods

  1. There is a need to provide details of what you have done for comparing the results of RCTs and RW study. Have you performed a meta-analysis or something like this for comparison? Please be elaborative in terms of the design of your research, data collection, data analysis process so on in here and add as much as you can to make your research repeatable by other researchers.

Response 8: We thank the reviewer for this useful suggestion. We adopted an innovative approach where each patient from the RW population was matched with a patient from the simulated RCT population in order to evaluate whether the outcome estimates reported in the RCTs were reproducible using data from the RW population sample. Regarding the quality and reproducibility of our research, we have provided details about simulations and methodological aspects in the Supplementary Methods file. Main results for RCT and RW studies were focused on the estimation and comparison of event rates for both the efficacy and safety outcomes. Although all the pivotal RCTs studies reported such event rates, they (unfortunately) did not report any precision measure, such as confidence interval, standard error or p-value, which accompanies each point estimate. For this reason, it was not possible to apply any statistical test (e.g. Cochran’s Q test like in a meta-analysis framework) to formally assess the heterogeneity among these estimates and therefore we just compares them in a descriptive way.

Results

  1. Please do not forget to use citations for any details comes through this section to find which finding belongs to which RCT or RW study.

Response 9: We have added the citations of the pivotal RCTs accordingly.

Discussion

  1. Line 42, please elaborate on it and describe those other RW studies. I suggest to compare your findings with those of RWs studies and systematic reviews in other settings. Therefore, do not reduce the level of your study thorough being compared with single RCTs.

Response 10: We thank the reviewer for this valuable comment. We point out at this stage that our results were generally consistent with similar RW studies. In an Australian study (ref. 18), clinical characteristics of RW patients treated with dabigatran and apixaban were fairly more comparable with those included in RCT than with rivaroxaban, similarly to the present study. In addition, the high burden of comorbidities or -on the contrary- a very low risk of stroke in RW population would have not qualified up to one third of RW population for inclusion in phase III RCT for stroke prevention in atrial fibrillation.  We have added it in the discussion section.

  1. Line 46, which recent Italian study?

Response 11: This refers to “Olimpieri PP, Di Lenarda A, Mammarella F, Gozzo L, Cirilli A, Cuomo M, et al. Non-vitamin K antagonist oral anticoagulation agents in patients with atrial fibrillation: Insights from Italian monitoring registries. IJC Hear Vasc. 2020; 26:100465”. We have added the number of the reference.

  1. Line 76 needs a citation, also.

Response 12: We have added the reference of the dabigatran Summary of Product Characteristics accordingly.

  1. Line 122 should be connected to line 121.

Response 13: We have connected line 121 with 122.

  1. Line 122, please describe those observational studies.

Response 14: We have described all the three mentioned observational studies.

  1. I can not understand why Methodshas been placed after the Discussion! A figure can help with summarising the research method. The description of the RW study should be separated from the RCTs.

Response 15: The body of the manuscript was prepared using the Microsoft Word Template provided by the Pharmaceuticals journal. Please see the following link: https://www.mdpi.com/journal/pharmaceuticals/instructions 

Conclusion

  1. It should contain implications of the study based on the findings. What would be the implications for education, management, policymaking etc?

Response 16: We thank the reviewer for this valuable comment. Although the implications of RW data for education and policymaking go beyond the scope of this manuscript, important implications for patient’s management apply for clinical perspectives. As suggested by the other two reviewers, we specified that this study provides reassurance to practicing physicians that DOAC use appears effective in stroke prevention and likely safer in RW population than in RCT population. These results may be related to a lower burden of comorbidities despite more advanced age in the RW population compared to pivotal RCT population. We have now added these considerations in the abstract and in the conclusions sections.

Round 2

Reviewer 2 Report

No further comments.

Reviewer 3 Report

Nothing more.

This manuscript is a resubmission of an earlier submission. The following is a list of the peer review reports and author responses from that submission.

Round 1

Reviewer 1 Report

This study aims to assess the external validity of the RCTs of the DOACs for the treatment of AF by comparing patients recruited in the phase-III RCTs to those treated with DOACs registered in the Southern Italian Local Health Unit (LHU) in the years 2013-2017.

The study question is interesting.

However, there are lot of weaknesses in the paper.

The main weakness is that the populations are completely different, which most probably can not be compensated with any kind of matching.

Also, the number of patients in the RW-setting is very limited, and this rises a question about selection bias – have all the available patients included in the RW cohort? Any skewness when employing the cohort would destroy the results.

First of all a suspicion rises when the authors report the risk of stroke 0.11% /pat.y in the matched RW-cohort compared to the RE-LY population 0.93% /pat.y. A suspicion of invalid analyses should have been raised also among the authors.

Regarding the discontinuation of a DOAC the authors should have taken into consideration patients planned for a cardioversion.

Low risk AF patients could have been stopped an OAC if sinus rhythm had been maintained for 1-3 months after a cardioversion.

It seems that the authors did not have laboratory data of the patients. However, CKD has been determined with history data, even “744 (10.5%) RW DOAC users had CKD; for 447 (60.0%) of these, CKD stage was known”. What was the status of renal function of the rest of about 6300 RW-patients?

In general, discussion about renal disease and dose reduction is irrelevant (lines 53-69) without the laboratory and weight data

The renal status was analysed in all the pivotal RCTs: RELY: Hijazi et al. Circulation 2014; ROCKET-AF: Fox et al. European Heart Journal 2011 and ENGAGE AF-TIMI 48: Bohula et al. Circulation 2016

Reference 12 should be replaced with the ESC 2020 version of AF Guidelines

The Reference 21 should not be used; the main article is challenging to reach, and is in Italian

The reference 22 (Schneeweiss et al.) is not a RW study, thus it is a meta-analysis of RCT:s (but not including the ENGAGE AF-TIMI 48)

Is the register data of the Italian National Health Service (NHS) validated? If so, this should be cited.

Reviewer 2 Report

Summary:

This is a comparative study performed to assess the validity of findings from the RCTs on DAOCs for NVAF by comparing them with real-world data from a claims database from southern Italy. The authors found a lower proportion of discontinuations for rivaroxaban and apixaban in the RW population, lower ischemic stroke rates in RW dabigatran users, and lesser bleeding rates in the real-world DOAC users compared to the RCT data. Compared to the RCT population, the baseline characteristics in the RW population had an equitable distribution of gender but a more elderly population.

Specific Comments:

  1. The analysis with the use of matching and utilizing the multivariable regression models to ensure confounding is limited was reasonable but can the authors describe why this discrepancy in the RW versus RCTs was seen with the age and gender specifically, although it is understood that high-risk factor patients including the higher presence of strokes/TIAs and CHADS2 scores may be more in the RCTs since they are generally more selective with respect to patients enrolled?
  2. Minor: “In the pivotal RCTs, 42,336 DOAC users with NVAF were overall enrolled.” Grammatical modification of the sentence needed.
  3. Minor: “In general, patients enrolled in ROCKET-AF study were more clinically compromised as compared with patients enrolled in other DOAC pivotal RCTs as well.” Can the authors clarify this sentence especially what they meant by “clinically compromised”?
  4. But for the above, this is a well-performed study and a nicely drafted manuscript.

Reviewer 3 Report

Dear authors,

The study is very interesting and very well designed.

Minor: 

  • standard doses of apixaban and rivaroxaban should be mentioned (other authors comment low doses, e.g., A Buchholz, et al. Initial apixaban dosing in patients with atrial fibrillation. Clin Cardiol. 2018 May; 41(5): 671–676.)
  • Fig 1b and 1c - the blue does not have the same meaning as in Fig 1a.